# Development of a Measurement Device for Micro Gas Flowrate Based on Laminar Flow Element with Micro-Curved Surface

**DOI:** 10.3390/mi15050660

**Published:** 2024-05-17

**Authors:** Zixuan Wang, Ya Xu, Tiejun Liu, Zhenwei Huang, Dailiang Xie

**Affiliations:** Zhejiang Key Laboratory of Flow Measurement Technology, China Jiliang University, Hangzhou 310018, China; p22020854123@cjlu.edu.cn (Z.W.);

**Keywords:** laminar flow sensing element, micro flow measurement, numerical modeling, linearity

## Abstract

The laminar flow meter (LFM) boasts several advantages such as no moving parts, a wide range ratio, high measurement accuracy, quick dynamic response, etc., and is a promising technology for micro gas flow measurement. In order to explore the influence of different curvature radii on curved surface gap LFM, three curved structures with different curvature radii were designed. The computational fluid dynamics method is applied to simulate the flow feature of three structures. The simulated velocity cloud and pressure distribution show that the larger the curvature radius, the more stable the flow of gas medium. The relationship between differential pressure and volume flow was obtained through the test within a flow range of 0~540 sccm. Regression analysis revealed that the volume flow measured by the curved surface LFM had a high linear relationship with the differential pressure. Experimental findings indicate that differential pressure of the structure with a curvature radius of 2 mm was greater than that of other two structures (curvature radius of 6 mm and 3 mm) at the same point. This indicates that adding the number of surfaces can effectively increase the pressure loss, so as to obtain a larger range ratio, but will increase the measurement error.

## 1. Introduction

Micro gas flow measurement technology is widely used in industries such as semiconductor processing [1], aerospace [2], automotive electronics [3], the chemical industry [4], and medical equipment [5]. These industries require high accuracy and short response times for micro gas flow measurement. Laminar flow technology has outstanding features in the measurement of micro gas flow owing to its advantages of no moving parts, a wide range ratio, accurate measurement, and fast response.

The research and development of laminar flow meters (LFMs) began in the middle of the 20th century. In 1957, Kreith et al. [6] analyzed the mechanism of internal resistance loss and flow characteristics for the flow condition of short capillary tube under a small Reynolds number, which made a great contribution to the research of laminar flow sensing technology and the subsequent development and application of LFM. For the traditional LFM, the actual measured pressure drop is not strictly linear with the variation in volume flow because of the entrance/exit effects. In order to reduce the proportion of nonlinear pressure loss to improve the measurement accuracy, a large capillary length-to-diameter ratio is required. National Institute of Standards and Technology (NIST) [7,8,9] chose quartz glass capillary as the laminar flow element (LFE), with a length of a few meters to tens of meters and an inner diameter of 300 μm or so. The capillary was wound around a cylinder with a diameter of about 100 mm, and the ratio of capillary length to diameter was more than 2000, which is very unfavorable for industrial and civil flowmeters. In order to reduce the inlet flow section of the nonlinear effect, Pena et al. [10] proposed a new disposition of LFE with three pressure points. The design idea was to take the pressure drop of the whole length of the capillary with micro flow, and to take the pressure drop of the second half of the capillary with large flows. In this way, it can effectively extend the measuring range and reduce the nonlinear error. However, this method is relatively complicated due to the valve switching system. Feng et al. researched and developed a flow standard device in metrology experiments [11] and found that when the diameter ratio was less than 500, the inlet section had a relatively large influence. Nuszkowski [12] used stainless steel wire bundles to make LFM, and its length-to-diameter ratio was nearly 3000. The consequences of a large length-to-diameter ratio are great pressure loss and difficulty to install. LFM performs better at low flow rates, so the traditional LFMs were chosen as a transfer standard of micro flow [13].

LFE is an important part of LFM, and the common structure forms are capillary, parallel plate, and flute. At present, LFE is mainly the capillary type. The appropriate number of capillary tubes can be selected according to the needs of the flow measurement. Shi [14] proposed an LFM with a parallel plate and utilized the gap formed between parallel plates as a flow channel, but there is a risk of gas leakage. The flute type [15,16] is not suitable for micro flow measurement because of its complex structure and high processing accuracy requirements. There are also special structures, such as the circular-segment-type [17]. This configuration of the installation structure will cause certain damage to the laminar flow, so the measurement accuracy cannot be guaranteed. Wang [18] et al. analyzed the flow resistance characteristics of two branches of the parallel pressure differential (PPD) laminar flow sensing element by digital simulation. The core idea of PPD [19] is to construct a double-channel structure with four capillary laminar flow components in cross symmetry. Different lengths are connected in series on each branch. The PPD laminar flow sensing technology can effectively reduce the entrance/exit effects. In recent years, Wang et al. [20] proposed a rectangle-gap-type LFE, in which two pressure taps are placed inward of the middle of the laminar flow channel to overcome the nonlinear effects. However, because the static pressure signal is measured from a capillary laminar flow channel instead of a pressure chamber, it pressure tap installation is difficult.

In summary, the previous research on the LFM was mainly focused on widening the limit of measurement and overcoming the nonlinear effects of sudden expansion and contraction. Thus, in this paper, a novel LFM with a curved surface gap is designed to expand the application of LFMs. The effects of curvature radius on flow measurement are analyzed and compared. The results show that the proposed curved surface gap structure has excellent linearity, and increasing the number of surfaces can effectively increase the pressure loss as a way to obtain a larger range ratio.

## 2. Working Principle of the Curved Surface Gap LFM

The LFM is a flow measurement device based on differential pressure measurement, which mainly includes an LFE and differential pressure sensor. The LFE is to ensure that the fluid passing through the flowmeter is in a laminar flow state.

In this paper, an LFM with a curved surface gap is proposed. The laminar flow channel consists of two parts, a straight section and a curved surface section. The curved surface section is 1/4 of a circle, and the laminar flow channel is a rectangular flow channel. The LFM is designed based on Hagan–Poissuet’s law. This law states that in a circular tube, under specific conditions including temperature and tube diameter, if the fluid flow is laminar, the flow rate $q_{v}$ is proportional to the pressure drop (Δp). This relationship can be expressed by the following equation:(1)$$q_{v}=\frac{\pi \Delta pd^{4}}{128\mu L}$$
where $q_{v}$ is the volume flow rate; d is the equivalent diameter of the round tube, m; L is the length of the round tube; μ is the dynamic viscosity of the fluid; and Δp is the differential pressure between the two ends of the round tube.

When considering the effect of gas compressibility, the volumetric flow rate is constantly changing along the flow direction, and is generally corrected by using a correction in the flow calculation [21]. The Hagen–Poissuet equation is satisfied between the pressure drop Δp and the volumetric flow rate $q_{v}$ over the length of the micrometric pipe section dz:(2)$$dq_{v}=\frac{\pi \Delta pd^{4}}{128\mu L}dz$$

Integrating Equation (2) gives the volume flow rate $q_{v1}$ at the inlet position:(3)$$q_{v1}=\frac{\pi d^{4}}{128L}\frac{\Delta p}{\mu }\frac{P_{1}+P_{2}}{2P_{2}}$$
where $P_{1}$ and $P_{2}$ are the capillary inlet and outlet pressures, respectively. The flow rate $q_{v1}$ at the inlet position is equivalent to taking the pressure in the middle of the tube section (i.e., the average pressure) for correction.

For parallel flat plate gap flow, the volumetric flow rate $q_{w}$ is:(4)$$q_{w}=\frac{WD^{3}}{12L}\frac{\Delta p}{\mu }\frac{P_{1}+P_{2}}{2P_{2}}$$
where *W* and *D* are the width and depth of parallel plates, respectively.

According to the ideal gas state equation, if each side of Equation (4) divided by $\left(P_{1}+P_{2}\right)/2P_{2},Q_{w}$ can be expressed by Equation (5):(5)$$Q_{w}=\frac{WD^{3}}{12L}\frac{\Delta p}{\mu }$$

The internal measurement principle of the LFM with the curved surface gap is as follows: When the fluid medium flows from the straight pipe section through the curved section, part of the potential energy is converted into kinetic energy. The flow velocity increases, resulting in increased friction losses along the way, and the static pressure continues to fall. According to Equation (5), the flow value can be obtained by measuring the pressure difference between the inlet and outlet of the pipe.

## 3. Physical Model and Simulation Analysis

### 3.1. Physical Model of the Flow Meter

Three curved surface structures with different curvature radii are designed. In order to ensure the full development of laminar flow, the pipe length should be long enough. The length of the entrance/exit of the LFM with the curved surface gap is calculated by the Shciller formula.
(6)$$L_{i}=L_{o}=Cd_{e}R_{e}$$
(7)$$de=\frac{4WD}{2\left(W+D\right)}$$
where $R_{e}$ is the Reynolds number; $L_{i}$ is the distance between the inlet of the laminar flow channel and the pressure extraction hole at the inlet end; $L_{o}$ is the distance between the laminar flow channel outlet and the pressure extraction hole at the outlet; C is the length coefficient of initial laminar flow section; and de is the equivalent diameter.

The main structural parameters of this LFM are determined by $q_{v}$ and Δp, which are calculated as follows.
(8)$$Re_{max}=\frac{\rho u_{max}de}{\mu }$$
(9)$$q_{max}=u_{max}WD$$
(10)$$\Delta p_{max}=\frac{12\mu L}{WD^{3}}\;q_{max}=\frac{12\mu L}{D^{2}}\;u_{max}$$
where $Re_{max}$ is the design maximum Reynolds number; $u_{max}$ is the design maximum average flow speed; $\Delta p_{max}$ is the design maximum pressure difference; and $q_{max}$ is the maximum volume flow rate. *L* is the distance between two pressure extraction holes in the laminar flow channel. The laminar flow channel of the curved surface gap LFM is a slit type, and its critical Reynolds number is smaller than the circular pipe, generally 1000 to 1100. Taking $Re_{max}$ as 560 and C as 0.02875, the parameters of the LFM with the curved surface gap are shown in Table 1. The structural diagrams are shown in Figure 1.

### 3.2. Simulation Parameter Setting

Air is chosen as the working medium for the numerical calculations, and its physical properties are density *ρ* = 1.225 kg/m^3^, dynamic viscosity *μ* = 1.7894 × 10^−5^ Pa·s. The boundary conditions are set to mass flow inlet and pressure outlet, and the wall material is set to aluminum, with a no-slip boundary condition. The linear and curved structures are given the same mass flow rate. For a given mass flow rate $q_{m}$, the Reynolds number $R_{e}$ is calculated as:(11)$$R_{e}=\frac{q_{m}de}{A\mu }$$
where *A* is the cross-sectional area of the rectangular pipe, in m^2^.

### 3.3. Analysis of Simulation Results

#### 3.3.1. Analysis of Velocity Distribution in the Flow Field

Given the same size of the inlet mass flow rate, *Re* = 100, the flow characters of three structures are analyzed. As shown in Figure 2, in the curved surface section, the phenomenon of stratified flow is more obvious. In the inlet and outlet parts, the velocity is almost zero near the upper and lower walls. The structure with a radius of curvature of 2 mm has a sudden change in velocity at the junction of straight and curved surface section. The structure with a radius of curvature of 6 mm has better stability at the interface between the front and back sections, and with a larger radius of curvature, the airflow has a longer buffer distance. It can be seen that the radius of curvature affects the gas flows steadily through the curved surface section.

#### 3.3.2. Internal Flow Loss

Figure 3 shows the static pressure variation of the four structures along the flow direction, and along the axis, the static pressure continues to drop. On the whole, the static pressure and the pressure loss of the curved structure is greater than that of the linear structure. In the linear structure, the main pressure drop comes from the friction loss inside the pipeline. The static pressure drops of the linear structure are nearly uniform, while the pressure loss of the curved structure is affected by its curvature radius. In the curved section (x = 15–39 mm), the fluid flows from the straight section to the curved surface section, flowing through 1/4 of the circle channel. Part of the potential energy is converted to kinetic energy, resulting in increasing velocity and increasing friction loss along the way [22]. As it passes through a semicircular curved channel, the fluid is cached because the radius of curvature increases. The range of static pressure drop is slightly smaller than that at the surface entrance. At the outlet, the fluid flows from the curved surface section to the straight section, and part of the dynamic pressure is converted to static pressure, so the static pressure drop amplitude is increased. The pressure loss of the structure with a radius of curvature of 2 mm is the largest. Because, under the same surface circumference (C = 12π), the smaller the curvature radius is, the more bending pipes the gas medium flow passes through.

## 4. Experimental Tests

### 4.1. Experimental Model

The experimental model was designed according to the principle of curved gap laminar flow sensing technology. The LFM with the curved surface gap was composed of an upper body and a lower body, which are connected by fastening screws. A thin sheet iron with a thickness of 0.2 mm was placed between the upper and lower bodies, and the laminar flow channel was the gap formed by folding the sheet iron. Sealing grooves ere provided on both sides of the laminar flow path to prevent gas leakage from both sides. The schematic structure is shown in Figure 4.

### 4.2. Experimental Test Method

The LFM with the curved surface gap was tested with a sonic nozzle gas flow standard device. The measurement range of the standard device was 0–1 slm, and the measurement uncertainty was ±0.25% (k = 2). The ConST221 intelligent digital differential pressure gauge with a differential pressure range of (−10~10) kPa and an accuracy level of 0.02 was used to measure the differential pressure. A digital pressure transmitter with a measuring range of (0~200) kPa and an accuracy class of 0.02 was used to measure the absolute pressure of the gas. Temperature was measured by a temperature sensor from 0 to 50 °C with an uncertainty of 0.05 °C.

The experimental setup is shown in Figure 5. The gas source was a high-pressure cylinder containing dry air. When the test device was running, the gas flowed out of the high-pressure cylinder, through the pressure reducing valve and the needle valve, and then through the curved surface gap LFM and, finally, into the standard device.

Affected by machining accuracy, the values of *W* and *D* may be biased. Therefore, the instrument coefficient K (determined by the calibration experiment) should be introduced during the experiment. At the same time, considering the influence of gas compressibility, it is necessary to calculate the volume flow rate under the working condition and calculate μ by measuring pressure and temperature. The flow chart of the whole flow measurement is shown in Figure 6.

### 4.3. Laminar Flow Sensor Calibration Experiment and Analysis

#### 4.3.1. Calibration of *K*

The ambient temperature was 25 °C and the atmospheric pressure was 95.8 kPa. Five calibration points were selected and set at 20%, 40%, 60%, 80%, and 100% of the full scale. The calibration data are recorded in Table 2.

The calibration coefficient *K* of the curved surface LFM was determined by the above experiment, and the calculation formula was as follows:(12)$$K=\frac{\sum_{1}^{n}\frac{Q_{s}}{Q_{m}}}{n}$$
*n* is the total number of measurements, which is 5 in this experiment.

According to the calibration data, the instrument coefficients of the radius of curvature of 6 mm, 3 mm, and 2 mm structures were 0.98981, 0.94506, and 0.90179, respectively. The final working volume flow rate was:(13)$$Q_{m}^{*}=KQ_{m}$$

#### 4.3.2. Three Kinds of Surface Structure Data Test

Using the calibrated value of K, the three structures were tested at different flow points. The flow points were set to 5%, 10%, 20%, 40%, 60%, 80%, and 100% of the full scale at each pressure. The same sonic nozzle and differential pressure gauge were used to test at the same flow points, and the data obtained are shown in Figure 7, Figure 8 and Figure 9.

As shown in Figure 7, the overall measurement error of the three structures was less than 1%, and the measurement error of micro flow points was higher than that of the large flow points. From the analysis of the principle of laminar flow sensing technology, the main reason for the measurement error of micro flow data may be manual reading error. Because the flow at the outlet of the flow channel caused the fluctuation of the internal flow field, the differential pressure signal was affected, so there was a certain error in the manual reading. The measurement error of the structure with a radius of curvature of 6 mm was minimum, because the flow had a longer buffer distance in the process of converting from a straight section to a curved surface section. Therefore, selecting a larger curvature radius can obtain a smoother flow field and effectively improve the measurement accuracy.

Experiments have verified the simulation conclusion: At the same flow point, the differential pressure of structures with radii of 2 mm and 3 mm is larger than that of structures with radii of 6 mm. The measured pressure difference and the standard flow value were analyzed, and the curve between them was fitted using the least square method. An excellent linearity was seen, as shown in Figure 8. The experimental results show that the radius of curvature has a certain influence on linearity. Under the same flow channel length, the linearity of the large curvature radius was better, indicating that increasing the curvature radius can effectively reduce the nonlinear influence caused by sudden expansion and contraction.

According to JJG 736-2012 [23] “Gas laminar Flow Sensor”, as shown in Figure 9, the repeatability of three curved surface gap LFM did not exceed 1/3 of the absolute value of the maximum allowable error (1%).

The test results show that the values of $\Delta p_{max}$ and $q_{max}$ basically conform to the expected design. The curved surface gap LFM can improve the measurement capability by increasing the number of parallel plates. The Alicat laminar gas mass flowmeter (flow range: 1~1000 sccm) had a product measurement deviation index of ±0.8%. The measurement deviation of this experimental model was less than ±0.9%, which is comparable to the established Alicat brand of main gas laminar flowmeters.

### 4.4. Uncertainty Analysis

Volumetric flow rate of laminar flow sensors in the standard state:(14)$$q=k*\Delta p*\xi *\left(\frac{T_{N}}{T_{m}}\right)*\left(\frac{p_{m}}{p_{N}}\right)$$
where: $p_{m},\;T_{m}$—absolute pressure (Pa) and thermodynamic temperature (K) of the gas at the inlet of the LFM; $p_{N},\;T_{N}$—absolute pressure (Pa) and thermodynamic temperature (K) in the standard state; ξ—viscosity correction coefficient, ξ = $\mu_{N}/\mu$; Δp—the average differential pressure value of the sensor (Pa); and *k*—sensor gauge coefficient.

Measured with a sonic nozzle gas standard device, the volume flow rate generated by the device in the standard state was:(15)$$q_{s1}=\left(\frac{T_{N}}{T_{s}}\right)*\left(\frac{p_{s}}{p_{N}}\right)*q_{s}$$
where: $q_{s1}$—the volume flow rate of the device under the standard state, m^3^/s; $p_{s},\;T_{s}$—absolute pressure (Pa) and thermodynamic temperature (K) of the gas in the device; and $q_{s}$—the volume flow rate of the device under working conditions, m^3^/s.

The instrument coefficient of the curved interstitial LFM (radius = 6 mm) was 0.98981, and the viscous coefficient was ξ = 0.997.

All the uncertainties of the curved surface gap LFM are shown in Table 3.

When the factor *k* = 2 was included, the extended uncertainty of the device was:(16)$$u_{cr}^{2}\left(q\right)=\overset{N}{\underset{i=1}{\sum }}c_{r}^{2}\left(x_{i}\right)*u_{r}^{2}\left(x_{i}\right)$$
(17)$$U\left(q\right)=k*u_{cr}^{2}\left(q\right)=2*0.258\%=0.517\%$$

## 5. Conclusions

In this paper, a curved surface gap LFM is proposed. Compared with the traditional plane structure design, the curved surface gap LFM has the following advantages.

(1)Introduction of curved surface gap structure: In contrast to traditional flat designs, the flowmeter incorporates a curved surface gap structure, effectively increasing the length of the flow channel while saving space. This innovative design offers a novel approach to addressing challenges in micro gas measurement.(2)Adjustable radius of curvature: By varying the radius of curvature, the length of the surface segment can be adjusted to meet measurement requirements across different ranges. This flexible design allows the curved surface gap laminar flowmeter to adapt to various applications and measurement ranges.(3)Comparable measurement accuracy to established brands: Experimental testing demonstrates that the curved surface gap laminar flowmeter, with a curvature radius of 6 mm, achieves a measurement deviation within ±0.8%, comparable to the established Alicat brand of main gas laminar flowmeters. The test results across the three structures consistently exhibit small measurement deviations, fulfilling the design requirement of ±1%.(4)Expanded measurement range and improved accuracy: The proposed design effectively expands the measurement range and enhances measurement accuracy. This study holds significant implications for the industrial application of laminar flowmeters flow metering technology.(5)Unaffected by deformation: Resolution of leakage and deformation issues in vane flowmeters: Vane flowmeters commonly encounter problems such as leakage and deformation. The study addresses these challenges by designing and testing three sample pieces using a sonic nozzle standard device, showcasing the superior measurement accuracy of the curved surface gap laminar flowmeter. The experimental findings confirm that the laminar flow element of the curved surface gap laminar flowmeter remains unaffected by deformation. Even without nonlinear correction, the differential pressure and flow rate maintain a strong linear relationship across all three structures, validating the rationality of the design.

## Figures and Tables

**Figure 1 micromachines-15-00660-f001:**
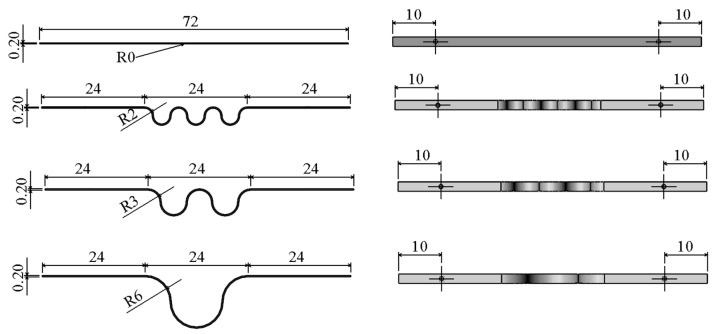
Four types of structural diagrams.

**Figure 2 micromachines-15-00660-f002:**
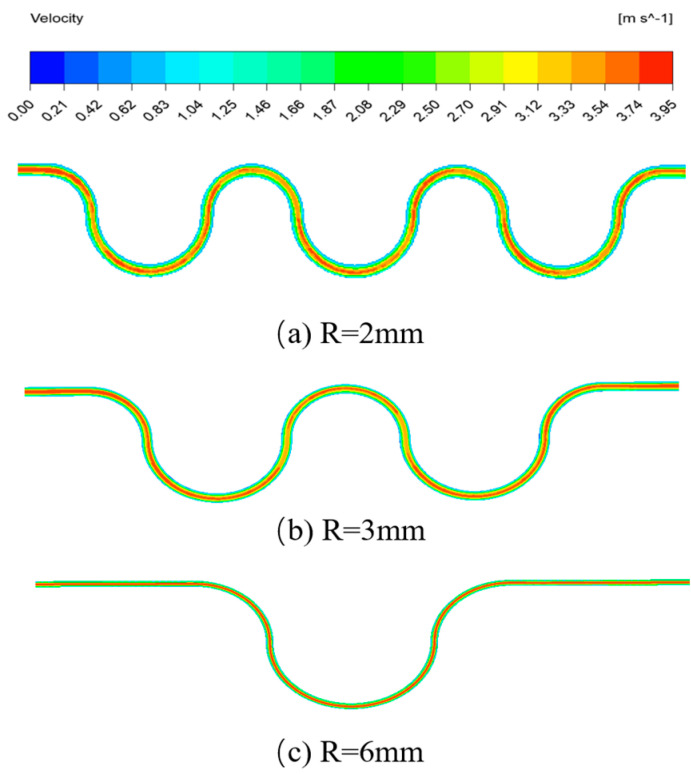
Velocity cloud image of three curved surface structures (*Re* = 100).

**Figure 3 micromachines-15-00660-f003:**
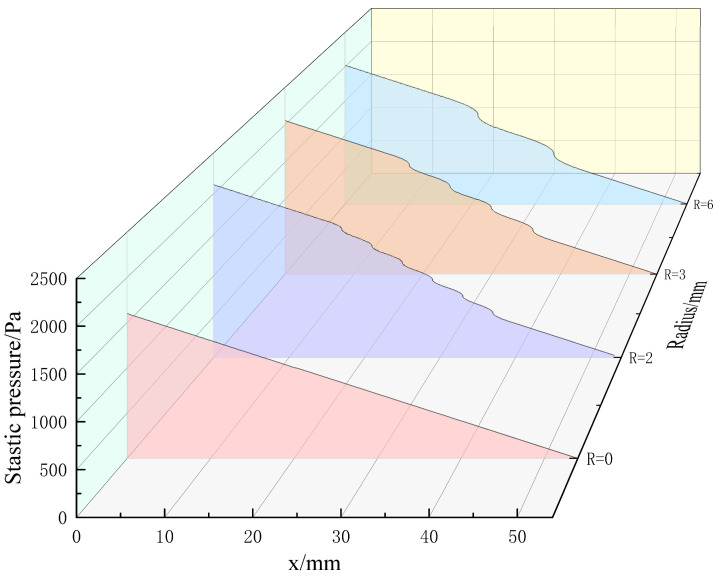
Axial static pressure curves of four structures (Re = 100).

**Figure 4 micromachines-15-00660-f004:**
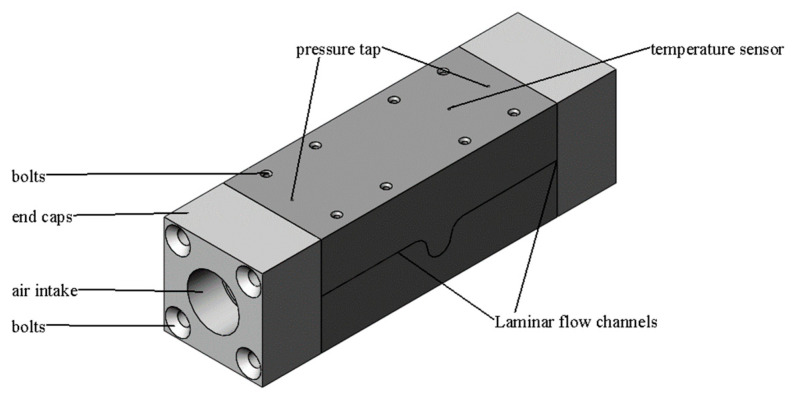
Schematic diagram of the structure of the curved interstitial LFM.

**Figure 5 micromachines-15-00660-f005:**
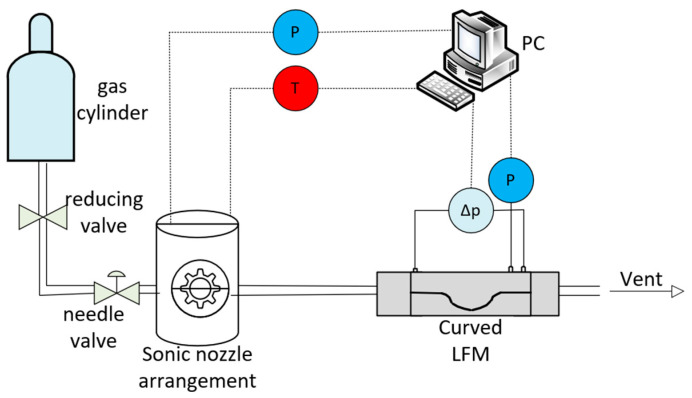
Structure of the test equipment.

**Figure 6 micromachines-15-00660-f006:**
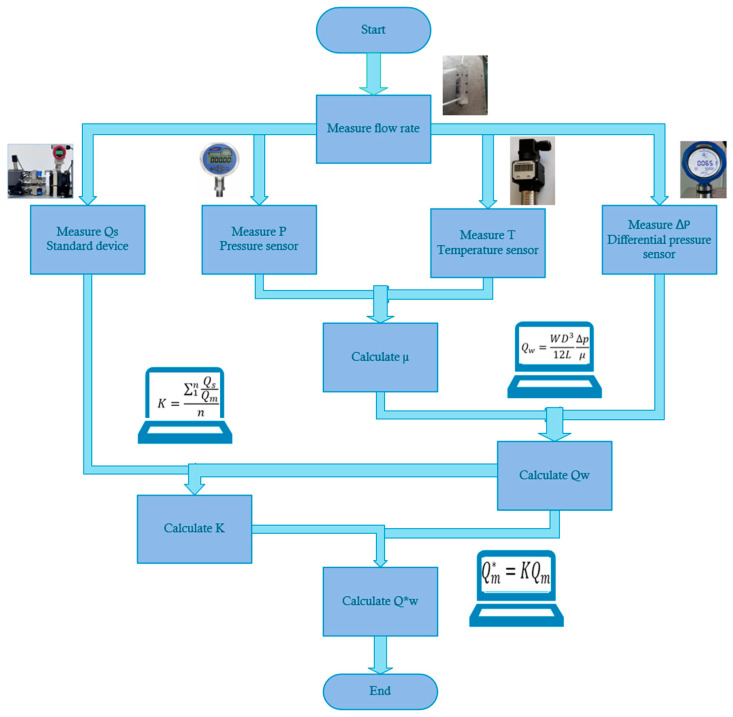
Flow diagram of flow measurement.

**Figure 7 micromachines-15-00660-f007:**
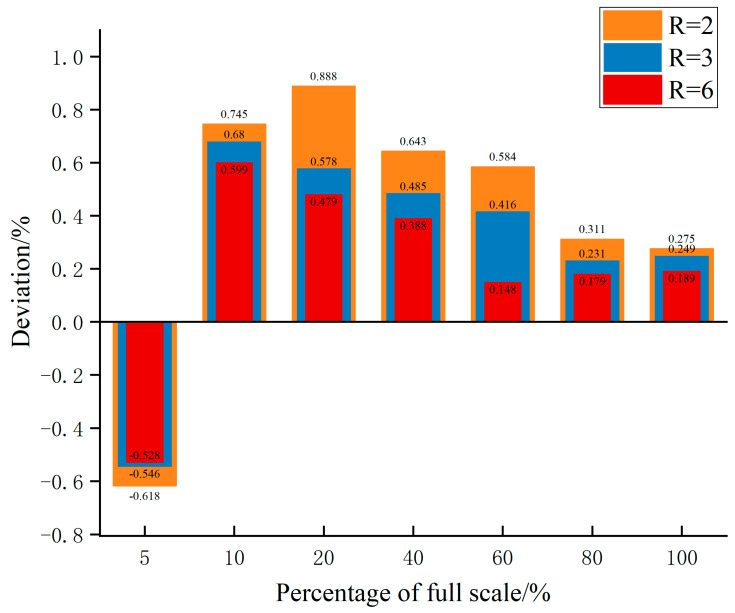
Tendency of the deviation at different flow points.

**Figure 8 micromachines-15-00660-f008:**
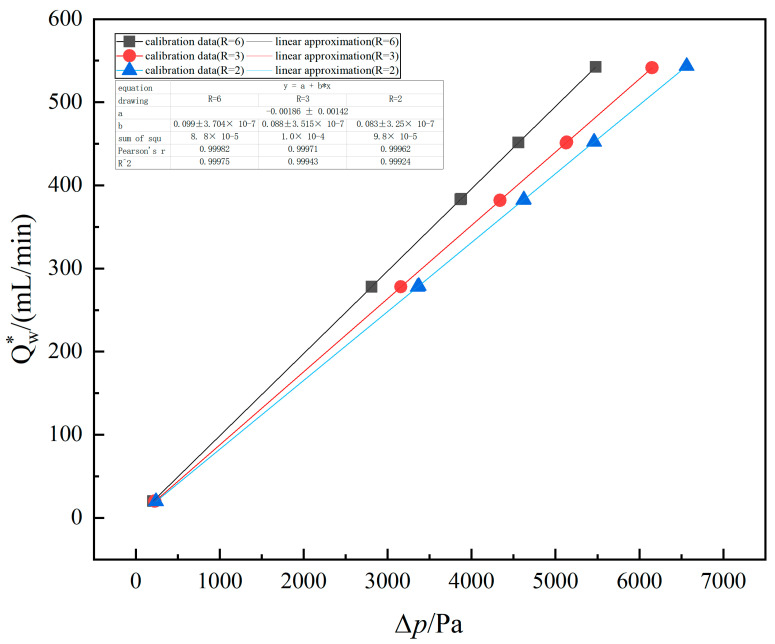
The fitting line of the test pressure difference and working volume flow value.

**Figure 9 micromachines-15-00660-f009:**
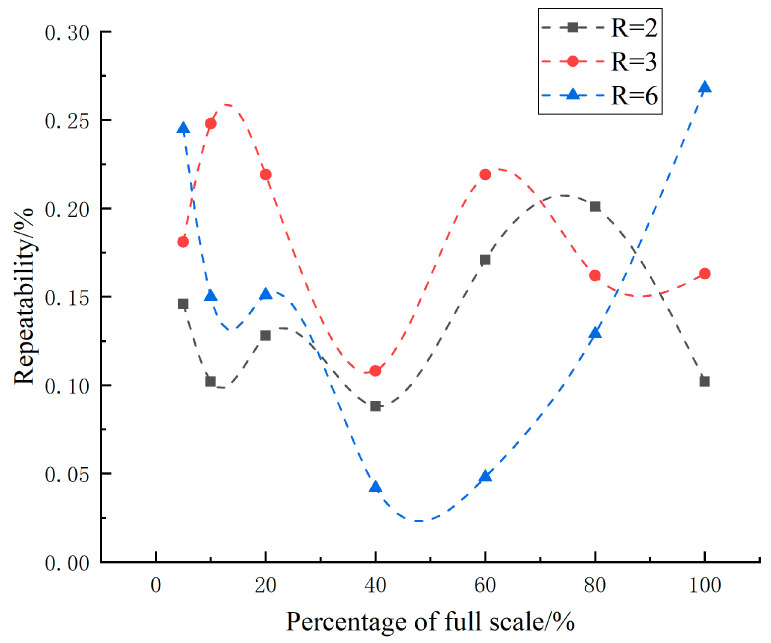
Tendency of the repeatability at different flow points.

**Table 1 micromachines-15-00660-t001:** Parameters design of the LFM.

Symbol	Units	Design Value
R	mm	0/2/3/6
W	mm	2
D	mm	0.2
$L_{i}$	mm	10
$L_{o}$	mm	10
L	mm	54
de	mm	0.36
$Re_{max}$	-	560
$u_{max}$	m/s	22.48
$q_{max}$	mL/min	539.52
$\Delta p_{max}$	Pa	6515.8

**Table 2 micromachines-15-00660-t002:** Calibration of the curved interstitial LFM (R = 6 mm).

No.	Calibration Point	Δp/Pa	$Q_{s}$/sccm	$Q_{m}$/sccm	*K*
1	20%	1027.3	108.1	108.98	0.9922
2	40%	2078.1	216.5	220.46	0.9813
3	60%	3080.1	324.1	326.76	0.9920
4	80%	3802.0	400.7	403.35	0.9934
5	100%	5144.6	540.2	545.77	0.9898

**Table 3 micromachines-15-00660-t003:** Uncertainties of the curved interstitial LFM.

No.	Source	Sensitivity	Relative Uncertainty %
1	$u_{r}\left(q_{s}\right)$	1	0.25
2	u_r_ ($T_{s}$)	−1	0.01
3	u_r_ ($T_{m}$)	1	0.01
4	u_r_ ($p_{s}$)	1	0.023
5	u_r_ ($p_{m}$)	−1	0.023
6	u_r_ (Δp)	1	0.046
7	u_r_ ($E_{r}$)	1	0.03

## Data Availability

The original contributions presented in the study are included in the article, further inquiries can be directed to the corresponding author.

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
