# Peer review of "Development of a Measurement Device for Micro Gas Flowrate Based on Laminar Flow Element with Micro-Curved Surface"

_micromachines, 2024, doi:10.3390/mi15050660_

Round 1

Reviewer 1 Report

Comments and Suggestions for Authors

The paper "Development of a measurement device for micro gas flowrate 2
based on laminar flow element with micro-curved surface" describes and effective methode for gas flow measurements. The sensor principle is described sufficiantly, however, some remarks:

1. Please show only data which is necessary for your argumentation and show it more clearly arranged, e.g. in graphs.

2. Please describe your set-up adn flow-diagramm in detail, otherwise people not familiar with your set-up won't understand it.

3. Please discuss your results, e.g. compare it with simmilar sensors or state-of-the-art.

Comments on the Quality of English Language

Although I am no native English speaker, I recommend to improve the English level...

Reviewer 2 Report

Comments and Suggestions for Authors

I have reviewed this manuscript entitled "Development of a measurement device for micro gas flowrate based on laminar flow element with micro-curved surface". The manuscript aims to establish the relationship between differential pressure and volume flow. The authors also did an effort in understanding the effects of curvature radius on flow measurement. The topic sounds scientific and interesting. However, the current version can be improved. I raised the following comments to help the authors improve the quality of the manuscript.

1) Please double check the line numbers in the manuscript. 92-94, 103-106, etc.

2) All of the acronym should be expanded first before using it as an acronym, such as LFM, LFE in the text.

3) Please indicate the basis for setting the parameters in Table 1. What is the engineering background of physical model?

4) Definitely, not all tables are required to demonstrate the research. Table 2 can be replaced by textual descriptions.

5) Figure 2: Please quantitatively provide the velocity distribution of the axis.

6) I do think the pictures in Figure 3 can be merged.

7) Page 6 Line nos. 198-208: please complement and enrich current statement with related literature: “Modeling of microflow during viscoelastic polymer flooding in heterogenous reservoirs of Daqing Oilfield(2022). J. Pet. Sci. Eng. 210, 110091”.

8) Please put enough emphasis on the points of novelty of the proposed study in your Conclusions.

9) References must be cited correctly and in a consistent manner, and to make sure they are cited within the manuscript.

10) Authors should check the manuscript for grammatical and language mistakes.

Comments on the Quality of English Language

Mentioned in the comments.

Round 2

Reviewer 1 Report

Comments and Suggestions for Authors

All questions have been answered carefully, hence I recommend to publish it.

Reviewer 2 Report

Comments and Suggestions for Authors

The authors have successfully responded to all of the reviewers' comments. Also, they edited the paper to answer all the concerns raised by the reviewers. My recommendation is publishable as is, and no additional work is needed.